# Excess Weight Impairs Oocyte Quality, as Reflected by mtDNA and BMP-15

**DOI:** 10.3390/cells13221872

**Published:** 2024-11-12

**Authors:** Emiliya Sigal, Maya Shavit, Yuval Atzmon, Nardin Aslih, Asaf Bilgory, Daniella Estrada, Mediea Michaeli, Nechama Rotfarb, Yasmin Shibli Abu-Raya, Shilhav Meisel-Sharon, Einat Shalom-Paz

**Affiliations:** 1IVF Unit, Department of Obstetrics and Gynecology, Hillel-Yaffe Medical Center, Hadera 3820302, Israel; maya.genel@gmail.com (M.S.); atzmony@gmail.com (Y.A.); nardin_aslih@yahoo.com (N.A.); asaf_bil@hotmail.com (A.B.); dra.danielaestradagarcia@gmail.com (D.E.); medeiam@hymc.gov.il (M.M.); nechami.ro@gmail.com (N.R.); yasmineshibli@gmail.com (Y.S.A.-R.); shilhav.ms@gmail.com (S.M.-S.); einats@hymc.gov.il (E.S.-P.); 2A Ruth and Bruce Rappaport School of Medicine, Technion-Israel Institute of Technology, Haifa 3525433, Israel

**Keywords:** obesity, body mass index, BMI, overweight, oocyte quality, BMP-15, HSPG2, mtDNA

## Abstract

This prospective, case-control study evaluated the impact of obesity on oocyte quality based on mtDNA expression in cumulus cells (CC), and on bone morphogenetic protein 15 (BMP-15) and heparan sulfate proteoglycan 2 (HSPG2) in follicular fluid (FF). It included women 18 to <40 years of age, divided according to BMI < 24.9 (Group 1, n = 28) and BMI > 25 (Group 2, n = 22). Demographics, treatment, and pregnancy outcomes were compared. The mtDNA in CC, BMP-15, HSPG2, the lipid profile, the hormonal profile, and C-reactive protein were evaluated in FF and in blood samples. The BMP-15 levels in FF and the mitochondrial DNA in CC were higher in Group 1 (38.8 ± 32.5 vs. 14.3 ± 10.8 ng/mL; *p* = 0.001 and 1.10 ± 0.3 vs. 0.87 ± 0.18-fold change; *p* = 0.016, respectively) than in Group 2. High-density lipoprotein levels in blood and FF were higher in Group 1 (62 ± 18 vs. 50 ± 12 mg/dL; *p* = 0.015 and 34 ± 26 vs. 20.9 ± 7.2 mg/dL; *p* = 0.05, respectively). Group 2 had higher blood C-reactive protein (7.1 ± 5.4 vs. 3.4 ± 4.3 mg/L; *p* = 0.015), FF (5.2 ± 3.8 vs. 1.5 ± 1.6 mg/L; *p* = 0.002) and low-density lipoprotein levels (91 ± 27 vs. 71 ± 22 mg/dL; *p* = 0.008) vs. Group 1. Group 1 demonstrated a trend toward a better clinical pregnancy rate (47.8% vs. 28.6%: *p* = 0.31) and frozen embryo transfer rate (69.2% vs. 53.8; *p* = 0.69). Higher BMI resulted in lower BMP-15 levels and reduced mtDNA expression, which reflect decreased oocyte quality in overweight women.

## 1. Introduction

Obesity is an increasing problem worldwide and is a risk factor for many diseases. It is defined as a BMI ≥ 30, whereas overweight is defined as a BMI ≥ 25 [1]. Overweight and obesity have negative effects on reproductive potential [2].

The effect of an increased BMI on female fertility has been evaluated extensively. Overweight and obese women present with a higher incidence of infertility, worse fertility treatment outcomes [3], decreased ovarian response, poorer oocyte quantity and quality, and lower-quality embryos [4,5,6,7].

With the aim of finding indirect markers for oocyte quality, it is acceptable to use follicular fluid (FF) and cumulus cells (CC), which serve as the microenvironment of the oocyte. Both CC and FF are critical determinants for oocytes and reflect their quality [8]. Evaluating FF and CC may help us understand the impact of obesity on oocyte quality without destroying the oocyte.

Mitochondria have a vital role in the metabolism of energy-containing compounds in the oocyte cytoplasm [9]. There is a significant relation between the amount of mitochondrial DNA (mtDNA) in the CC surrounding an oocyte and embryo quality [10]. Significantly higher mtDNA copy numbers are associated with good-quality embryos [10] and embryo implantation [11].

Proteins secreted from the oocyte into the FF are used as biomarkers for oocyte quality. Bone morphogenetic protein 15 (BMP-15) is secreted by the oocyte and acts within the ovarian follicle to control the function of the oocyte’s neighboring granulosa and CC to regulate folliculogenesis and fecundity [12,13]. Adequate amounts of BMP-15 secreted in FF are critical for female fertility [14]. The BMP-15 level in FF appears to be a factor in predicting oocyte quality and subsequent embryo development [15].

Another protein reflecting oocyte quality is heparan sulfate proteoglycan 2 (HSPG2), a key factor in proper follicular function and implantation. This protein is produced by granulosa cells and is secreted into the FF. It serves as the major estrogen-binding protein in the FF [16,17].

CRP is an important inflammatory marker. A strong correlation was shown between serum and FF CRP levels and a higher BMI [18].

Cholesterol is a precursor to steroid hormones, playing a pivotal role in the maturation of ovarian follicles [19]. In the follicular stage, the thecal cells absorb cholesterol from high-density lipoproteins (HDL) in the blood and produce androgens. These substrates cross the basal membrane and stimulate estrogen production by granulosa cells within the developing follicle [20]. HDL transports cholesterol to the ovarian follicle and is the main source of cholesterol substrate for steroidogenesis in the pre-ovulatory period [21,22].

This study evaluated the impact of obesity on oocyte quality using three main factors that reflect oocyte quality: mtDNA levels in CC, and levels of BMP-15 and HSPG2 in FF.

## 2. Materials and Methods

### 2.1. Patients and Samples

The study was conducted in the in vitro fertilization (IVF) unit from February 2022 to 2023. Women younger than age 40 years, undergoing IVF/IVF-intracytoplasmic sperm injection (ICSI) cycles, with four or more follicles at the time of ultrasound-guided transvaginal ovum pick-up (OPU) were enrolled. The patients were allocated into two BMI groups (<24.9 and >25) based on criteria established by the World Health Organization to evaluate distinctly different groups [1]. Demographic information was recorded from the electronic medical records. Women who had undergone fertility preservation or had comorbidities were excluded.

### 2.2. Treatment Protocol

Ovarian stimulation was initiated using 150–450 IU recombinant follicle-stimulating hormone (FSH) (Pergoveris or Gonal-F, Merck-Serono, Darmstadt, Germany) or hMG (Menopur, Ferring Pharmaceuticals, Lausanne, Switzerland), with age-specific adjustments based on basal hormone levels, antral follicular count, and BMI. At each follow-up visit, progesterone, estrogen (E2), and luteinizing hormone (LH) levels were measured, including on the day of hCG injection (Ovitrelle, Merck-Serono, Darmstadt, Germany). When at least two follicles with a mean diameter of 17–18 mm were detected, ovulation was induced, and oocyte retrieval was performed 36 h later.

Following oocyte retrieval, either IVF or ICSI was conducted. The Known Implantation Data (KID) score and morphology were used to grade embryos [23]. A top-quality embryo was defined as one with 4–5 cells on day two or >6 equal-size blastomeres on day 3, ≤20% fragmentation, and no multinucleate cells [24]. The remaining top-quality embryos were vitrified and transferred in the subsequent frozen embryo transfer (FET) cycle if pregnancy did not occur.

Twelve days after embryo transfer, the beta-hCG level was assessed. A beta-hCG > 25 mIU/mL was considered a chemical pregnancy. Clinical pregnancy was confirmed when an ultrasound detected a gestational sac with a fetal heartbeat at 7 weeks of gestation.

### 2.3. Blood and Follicular Fluid Collection and Cumulus Cell Isolation

On the day of oocyte retrieval, blood samples were drawn. After removing the oocyte, FF was collected and both were analyzed for hormone levels, including estradiol, progesterone, CRP, and the lipid profile. Before ICSI, oocytes were denuded from CC using hyaluronic acid and kept in 500 µL of physiological serum in each plate.

The FF of all aspirated follicles > 14 mm was collected and centrifugated at 1500× *g* for 6 min at room temperature. The supernatant was transferred to tubes and frozen at −80 °C. Following mechanical peeling from the oocytes using hyaluronic acid, CC were collected from plates by centrifugation at 4000× *g* for 6 min. The supernatant was removed, and the pellets were immediately frozen at −80 °C until DNA extraction.

### 2.4. Measuring Hormone and Lipid Levels in Plasma and Follicular Fluid

Progesterone and estradiol levels in the plasma and follicular fluid were measured using an electrochemiluminescence immunoassay on a Roche Cobas 8000, e801. The analytical sensitivity L (0.05 ng/mL) and the levels of LH and FSH in the plasma and follicular fluid were measured using an electrochemiluminescence immunoassay on a Cobas 8000, e801 (Roche Diagnostics, Basel, Switzerland). The analytical sensitivity LOD was 0.1 mIU/mL. The lipid profile in the plasma and follicular fluid was analyzed on a Cobas C701 (Roche Diagnostics, Basel, Switzerland).

### 2.5. DNA Extraction

DNA was extracted from isolated fractions of CC using the QIAamp DNA Mini Kit (QIAGEN, Hilden, Germany), according to the manufacturer’s protocol. The DNA concentration was measured using a NanoDrop spectrophotometer (Thermo Scientific, Waltham, MA, USA).

### 2.6. Quantification of mtDNA

The mean mtDNA copy number in the CC was determined using a Real-Time Quantitative Reverse Transcription Polymerase Chain Reaction (qRT-PCR) using SYBR^®^ Green master mix (Roche Diagnostics, Basel, Switzerland) in a 20 µL reaction volume containing a final concentration of 0.5 µM of each gene-specific primer and 3 µL of template. The pairs of primers selected were MT-CO1 (mtDNA nucleotide positions 7017–7036 and 7205–7224) to quantitate the mtDNA and 36B4 (36B4f, CAGCAAGTGGGAAGGTGTAATCC; 36B4r, CCCATTCTATCATCAACGGGTACAA) to quantitate the nuclear DNA (nDNA) in the CC. In each qRT-PCR experiment, we used a negative control sample composed of SYBR mix and water instead of a template. The qRT-PCR reactions were performed using a CFX96 detection amplifier (Bio-Rad, Hercules, CA, USA). The reactions were performed as follows: initial activation of DNA polymerase for 5 min and 40 cycles consisting of 15 s at 95 °C and 1 min at 58 °C. Analyses were carried out using the Bio-Rad CFX Manager 2.1 software package (Bio-Rad, Hercules, CA, USA) to generate the standard curve for each plate and to compute the mtDNA and nDNA values. The average CC mtDNA was determined by calculating the ratio between the mtDNA copy number and the nuclear DNA copy number (2^ΔΔCT^ equation) [25].

### 2.7. ELISA (Enzyme-Linked Immunosorbent Assay)

#### 2.7.1. BMP-15

The BMP-15 concentration in the FF was analyzed using an ELISA kit (Novus Biologicals, Toronto, ON, Canada), according to the manufacturer’s protocol. All the samples were assayed at the optimal dilution (1:10) and then assayed in a 96-well plate.

#### 2.7.2. HSPG2

The HSPG2 concentration in the FF was analyzed using the Human HSPG2 SimpleStep ELISA kit (Abcam, Cambridge, UK), according to the manufacturer’s protocol. The samples were diluted at 1:200 using the sample diluent provided.

The optical density for both kits was quantified using an SQ2 ELISA processor (Aesku Systems, Wendelscheim, Germany) at 450 nm.

The inter- and intra-assay coefficients of variation were <15%.

### 2.8. Statistical Analysis

The data were analyzed using SPSS-22.0 for Windows (IBM Corp., Armonk, NY, USA). All continuous variables were described with means and standard deviations and were compared using the Student’s *t*-test. The normality was analyzed using the Shapiro-Wilk test. The categorical variables were analyzed using a chi-squared or Fisher’s exact test. For results that were significant or showed a statistical trend in univariate analysis, multivariable model analysis was performed. Linear regression was used to evaluate the effect of the patients’ characteristics on the mtDNA and on the BMP-15 in the FF. A receiver operating characteristic (ROC) Youden index statistics curve was used to predict the mtDNA and BMP-15 cutoffs for distinguishing between BMI Groups 1, and the correlations were analyzed using Pearson or Spearman tests. A *p*-value < 0.05 was considered statistically significant. All statistical tests were two-tailed.

The sample size was calculated based on the mtDNA concentration in the CC [10,26] using a large effect size (0.8) and a power level of 85%, at a significance level of 0.05. At least 48 participants (24 in each group) were determined as sufficient to detect meaningful differences or associations in the primary outcome of the impact of overweight on oocyte quality based on the mtDNA expression in the CC and the levels of BMP-15, and on HSPG2 in the FF.

## 3. Results

A total of 50 women were enrolled: 28 with BMI < 24.9 (Group 1) and 22 with BMI > 25 (Group 2). Age, infertility causes, and other baseline clinical characteristics were similar between the groups (Table 1). All chronic diseases, including diabetes and hypothyroidism, were medically treated and balanced.

Table 2 presents the baseline hormonal profiles and treatment outcome parameters after ovarian stimulation that were compared between the groups. The antral follicle count, the number of treatment days, the gonadotropin dose, the treatment protocol, the gonadotropin and ovulation induction treatment, the endometrial thickness, the number and quality of oocytes collected, the fertilization rate, and the KID scores were all comparable. The baseline hormonal profile was comparable between the groups, except that Group 1 had higher FSH and E2 levels on trigger day, without clinical significance.

There was no significant difference in pregnancy rates between the groups, but there was a trend toward higher clinical pregnancy rates in Group 1 compared with Group 2 in fresh cycles (47.8% vs. 28.6%: *p* = 0.31) as well as in FET (69.2% vs. 53.8%; *p* = 0.69).

Table 3 presents the chemical parameters in the blood and follicular fluid. The CRP levels in the blood and in the FF and the low-density lipoprotein (LDL) in the blood were significantly higher in Group 2 compared to Group 1 (blood 7.1 ± 5.4 vs. 3.4 ± 4.3, *p* = 0.015, FF 5.2 ± 3.8 vs. 1.5 ± 1.6; *p* = 0.002; and FET 91 ± 27 vs. 71 ± 22, *p* = 0.008, respectively). However, the high-density lipoprotein (HDL) levels in Group 2 were significantly lower in the blood (50 ± 12 vs. 62 ± 18; *p* = 0.015) and in the FF (20.9 ± 7.2 vs. 34 ± 26; *p* = 0.05).

We found higher BMP-15 levels (Table 3 and Figure 1) and higher expression of mtDNA (Table 3 and Figure 2) (34.8 ± 26.2 vs. 15.1 ± 10.5 ng/mL; *p* = 0.006; 1.10 ± 0.3 vs. 0.87 ± 0.18, *p* = 0.016; respectively) in Group 1 compared to Group 2. The HSPG2 levels in Groups 1 and 2 were comparable (360.8 ± 119.2 vs. 440.6 ± 271 ng/mL; *p* = 0.264, respectively).

We also compared the group with BMI < 25 (28 patients) to those with BMI > 30 (12 patients) to distinguish between similar BMI values and found comparable results. Since the group with BMI > 30 was small, we decided to analyze the entire cohort based on BMI groups < 24.9 and >25.

The BMP-15 and mtDNA expression levels were positively correlated (r = 0.608; *p* = 0.001). The HSPG2 levels were not significantly correlated to BMP-15 and mtDNA. The BMP-15 was correlated with blood LDL levels (r = −0.386; *p* = 0.018) and the mtDNA was significantly correlated with maternal age (r = 0.580; *p* = 0.01) and the blood HDL levels (r = 0.547; *p* = 0.02). There was no significant correlation between the BMP-15, HSPG2, and mtDNA with the M2 oocytes and fertilization rate.

Multivariate linear regression analysis for the mtDNA demonstrated a significant predictive power of age (0.001) and E2 on trigger day (0.037) (F (3.68), *p* < 0.05), with the adjusted R^2^ = 0.51.

ROC analysis was used to predict the cutoff of the mtDNA and BMP-15 in relation to the BMI groups. We found that a BMP-15 level < 16.9 ng/mL correlated with a BMI > 25 (AUC 0.778, 95% CI [0.641, 0.914], *p* < 0.001) and a relative mtDNA level of <0.91 was correlated with a BMI > 25 (AUC 0.719, 95% CI [0.549, 0.890], *p* = 0.012). A cutoff value of BMP-15 > 16.9 was related to higher positive pregnancy test rates (30.4% vs. 69.6%, *p* = 0.334).

## 4. Discussion

FF performs ultrafiltration of the blood and represents various metabolic changes that have a strong effect on the oocyte. This study evaluated the impact of BMI level on oocyte quality by evaluating the amounts of two proteins in FF and the mitochondrial DNA in CC.

We found significantly higher BMP-15 levels in the FF of women with a normal BMI compared to those who were overweight as well as higher expression of mitochondrial DNA in CC. The HSPG2 levels were comparable between the groups.

Luke et al. [5] and Bellver et al. [27] both hypothesized that obesity is associated with female infertility and poor artificial reproductive technology (ART) outcomes due to impaired oocyte quality. Evaluating oocyte quality directly damages the oocyte and prevents its use, which reduces ART outcomes. Therefore, FF markers, which represent the environment of the oocyte, can serve as reflectors of the oocyte’s condition. Proteins secreted into the FF are known to correlate with oocyte quality. BMP-15 is a protein synthesized by the oocyte and is essential for folliculogenesis, the growth and maturation of ovarian follicles, the regulation of GC sensitivity to FSH, GC activities, and factors predicting oocyte quality and female fertility [15,22,23]. Higher levels of BMP-15 in FF were significantly correlated with a higher fertilization rate, better cleavage and with quality embryo development [15,28,29].

Different studies showed that higher BMP-15 levels were related to better oocyte quality [23] and better clinical results [15,25]. The BMP-15 levels ranged from 58 pg/mL to 278 ng/mL using different studies and different kits [25,27,28,29,30,31,32]. The present study found significantly higher levels of BMP-15 in the FF of women with a BMI < 24.9 compared to women with a BMI > 25. This may reflect higher oocyte quality in women with a normal BMI. Additionally, using ROC analysis, we found that a BMP-15 cutoff level of 16.9 ng/mL allows one to distinguish between a normal and increased BMI. Moreover, in agreement with the above studies, we found that normal-weight women had a BMP-15 level above the threshold, which correlated to a higher pregnancy rate.

The present study evaluated HSPG2 (a protein recognized as a key element in follicular function) in the FF. It is produced by granulosa cells and secreted by them into the FF [30]. HSPG2 expression increases during early follicular growth and enables the binding of various growth factors, which are essential for follicular growth and differentiation [31].

Few studies have evaluated the function of HSPG2 in reproduction. HSPG2 increases with gonadotrophin stimulation and mirrors follicular estrogen production. It was identified as the main estradiol-binding protein in FF and supports multiple biological activities that are relevant to embryonic development, such as cell adhesion, growth factor binding, and apoptosis regulation [30].

The evidence regarding HSPG2 levels is inconclusive. Bayasula et al. showed that the quantity of HSPG2 in the FF of fertilized oocytes was greater than in non-fertilized oocytes from the same patient. Furthermore, low levels of HSPG2 were observed in the FF of patients with PCOS [16]. In contrast, one study found that in patients with PCOS and normal ovulation, lower HSPG2 expression levels in CC were related to better-controlled ovarian hyperstimulation outcomes [33], and another study reported that lower HSPG2 expression in normo-ovulatory women was associated with more mature oocytes, transplantable embryos, and good quality embryos [32].

The present study demonstrated comparable levels of HSPG2 between the two BMI groups. This might be because it is the main estradiol-binding protein and was measured in stimulated cycles when estradiol levels were elevated and comparable between the groups. In addition, the small sample size might have prevented us from detecting differences.

mtDNA is a well-established marker that represents oocyte quality. Oocyte quality was shown to be influenced by the mtDNA content in the oocyte and in CC in humans and other species [34,35,36]. CC mitochondria are known to be central agents in the metabolic pathways and are directly involved in the establishment of oocyte quality during oogenesis [37]. Oocyte quality is related to the oocyte mtDNA copy number in humans, and the mtDNA copy number of an oocyte is linked to that of the corresponding CC [10,33]. Mobarak et al. [38] suggested that transferring mtDNA from several oocytes into one oocyte may improve the outcome of the transferred oocyte.

The present study found higher mtDNA expression in the group with a normal BMI compared to the BMI > 25 group. Our results agree with those of Desquiret-Dumas et al. who reported that patients with a higher BMI had lower CC mtDNA copy numbers than those with a lower BMI [10].

### 4.1. Relation of Age to BMP-15 and mtDNA

There is a negative correlation between maternal age and fertility [39]. It is well-established that oocyte quality declines with age [40]. In accordance with a study by Hashim et al. [41], which revealed that there was no significant correlation between age and serum and FF BMP-15, the present study also did not find any correlation between age and BMP-15 levels, probably due to the small, young, and homogeneous sample of patients (31.9 ± 5.4 vs. 29.1 ± 5.9 years of age).

Surprisingly, we did not find a difference in mtDNA expression related to women’s age. Most of the women in our study were younger than age 35, so it is difficult to draw any conclusions about the effect of age on oocyte quality. However, according to Martínez-Moro et al. [42], the relative amount of mtDNA in CC was associated with age, and most studies using animals and humans have shown that mtDNA copy numbers are relatively lower in older oocytes when compared with oocytes from young women [43,44,45]. In addition, in our study, most women over 35 had a lower BMI. Age and BMI affect the quality of eggs in older women, which might balance each other and improve the mtDNA concentration in older, slimmer women, which was expressed by a trend of higher mtDNA for the small group of women over 35 years.

High CRP levels are characteristic of increased inflammation and oxidative stress [46], which have a strong influence on oocyte development and subsequent embryo quality [47]. The current study found significantly higher CRP levels in the blood and follicular fluid in the BMI > 25 group compared to the low BMI group. Thus, our results support those of Schon et al. [48], Raviv et al. [49], and Haikin Herzberger et al. [50], who reported significantly higher CRP levels in FF from obese women compared with those of normal weight.

CRP as a marker of inflammation may explain the poorer ART outcomes in overweight and obese patients [50,51,52]. A high BMI has been related to increased CRP levels regardless of ovarian stimulation, which suggests an increased basal inflammatory state in overweight individuals. The association between high, circulating preconception CRP and poor ART outcomes could be due to a detrimental effect of CRP on the ovarian milieu and endometrial receptivity. It has been suggested that an excessive inflammatory response in decidual cells reduces the window of receptivity [53].

### 4.2. Lipid Profile

It has been reported that the quality of oocytes in obese women has been compromised. The compromised oocyte quality may be due to the imbalance or excess of lipids during oocyte development. Generally, lipids are mainly stored in the form of lipid droplets and are an important source of energy metabolism. Similarly, lipids are also essential signaling molecules involved in various biological cascades of oocyte maturation, growth, and oocyte competence acquisition [54]

Though lipid metabolites are indispensable, long-term exposure to a high-fat environment will induce irreversible damage to follicular cells and oocyte meiosis [55]. Some lipid metabolites are also important regulators of oocyte meiosis and maturation. On the other hand, excessive lipid accumulation will cause serious damage to ovarian reproductive function by inducing ovarian oxidative stress and inflammation [55].

In the present study, HDL levels in the blood and in FF were significantly higher in patients with a BMI < 25 compared to patients with a higher BMI. A review by Arias et al. suggested that FF HDL plays a role in maintaining balanced cholesterol levels in developing oocytes by removing excess cholesterol from the plasma membrane. Disruptions in HDL metabolism could impair FF HDL function, leading to infertility due to dysfunctional and unstable oocytes [56]. The BMI-dependent concentration of HDL in the serum was significantly associated with those of the FF concentrations, which suggests that changes in serum cholesterol levels could influence oocyte quality. Notably, it has been hypothesized that the combined impact of FF HDL plays a protective role in oocyte health and embryo development by reducing embryo fragmentation [7]. Our results agree with those of Valckx et al. [7], who revealed that the concentration of serum HDL cholesterol was significantly associated with FF concentrations, indicating that FF HDL cholesterol is vulnerable to serum cholesterol changes and could, therefore, influence the quality of the cumulus–oocyte complex.

### 4.3. Treatment Outcomes

Due to the relatively small sample size and young age of the patients in both groups, we could not demonstrate a significant difference in the pregnancy rate according to BMI group. However, there was a trend toward better results in the group with BMI < 25, which might reflect better oocyte quality in the women with a BMI in the normal range. This finding agrees with those of Luke et al. [5] and Sermondade et al. [5,57], who showed that the likelihood of achieving a live birth decreases with increasing BMI.

Additionally, we found that a cutoff value of 16.9 ng/mL of BMP-15 distinguished between BMI values of <24.9 or ≥25. This cutoff revealed a trend toward better pregnancy rates (30.4% vs. 69.6%, *p* = 0.334). This might become more apparent with a larger sample size.

### 4.4. Limitations

The major limitation of this study was the small sample size. As a result, we were unable to demonstrate a significant difference in pregnancy rates. The difference in the BMI between the groups was not very pronounced. In addition, we cannot report the cumulative pregnancy rate because not all the frozen embryos have been transferred yet.

## 5. Conclusions

The findings presented here demonstrate the relationship between an increased BMI and lower levels of BMP-15 and reduced expression of mtDNA, which reflect decreased oocyte quality in overweight women. We did not find a significant difference in HSPG2 levels between the groups. We assume that the higher pregnancy rate in the lower BMI group is related to better oocyte quality, but additional studies are needed to confirm this hypothesis. Further insights might be obtained using a larger sample size. Our recommendations to our patients include losing weight and improving their lifestyle in order to overcome the negative effects of these impaired quality markers.

## Figures and Tables

**Figure 1 cells-13-01872-f001:**
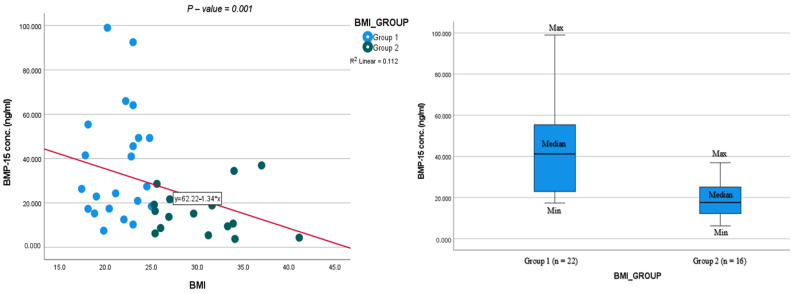
Scatterplot and boxplot correlation of BMP-15 and BMI.

**Figure 2 cells-13-01872-f002:**
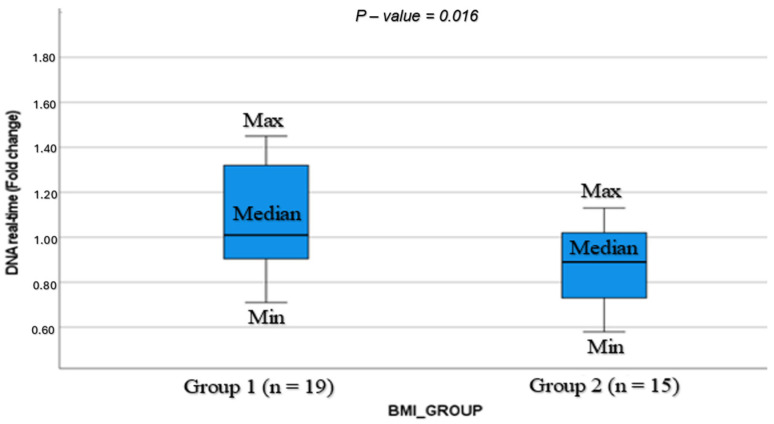
Boxplot showing the fold change expression of mtDNA according to BMI.

**Table 1 cells-13-01872-t001:** Clinical characteristics of the study population according to BMI.

Characteristic *	BMI < 24.9 n = 28	BMI > 25 n = 22	*p*-Value	95% CI
Age (y) mean ± SD	31.86 ± 5.42	29.14 ± 5.88	0.09	
BMI (kg/m^2^)	21.44 ± 2.19	31.13 ± 4.69	<0.001	[−11.7, −7.7]
Chronic disease	12 (41.3%)	10 (47.6%)	0.775	
Medications	9 (31%)	8 (38.1%)	0.76	
Smoking	10 (37%)	4 (21%)	0.33	
Gravidity	0.97 ± 1.43	0.62 ± 1.12	0.34	
Parity	0.48 ± 0.63	0.29 ± 0.46	0.21	
Cesarean section	3 (10.3%)	3 (14.3%)	0.69	
Miscarriage				
1	6 (20.7%)	2 (9.5%)	0.3	
2 or more	2 (6.9%)	2 (9.5%)	0.8	
Cause of infertility				
Male factor	17 (58.3%)	10 (47.6%)	0.56	
Mechanical factor	3 (10.3%)	2 (9.5%)	1.0	
Unexplained	9 (31%)	5 (23.8%)	0.75	
Anovulation	0	4 (19%)	0.026	

* Values are mean ± SD or n (%).

**Table 2 cells-13-01872-t002:** Hormonal profile and treatment outcomes based on BMI.

Characteristic *	BMI < 24.9	BMI > 25	*p*-Value	95% CI
Hormonal profile				
FSH baseline (IU/L)	7.88 ± 2.2	6.42 ± 1.91	0.019	[0.25, 2.7]
LH baseline (IU/L)	6.2 ± 2.9	4.6 ± 2.4	0.46	
E2 baseline pg/mL)	30 ± 75.8	40 ± 77.5	0.11	[−14, 1216]
E2 on trigger day (pg/mL)	2469 ± 1186	1868 ± 878	0.055	
E2 on ovum pick-up day (pg/mL)	1399 ± 1123	1157 ± 525	0.38	
Antral follicle count	16 ± 9.1	18 ± 9.3	0.475	
Total days of treatment	10.6 ± 2.5	9.7 ± 1.8	0.17	
Total gonadotropin dose (units)	2833 ± 1515	2415 ± 1195	0.29	
Treatment protocol				
Antagonist	24 (82.1%)	19 (90.5%)	0.35	
Other	5 (17.9%)	2 (9.5%)		
Gonadotropin treatment			0.581	
Menopur	17 (58.6%)	13 (61.9%)		
Pergoveris	11 (37.9%)	6 (28.6%)		
Gonal F	1 (3.4%)	2 (9.5%)		
Ovulation induction			0.33	
Ovitrelle	5 (17.2%)	1 (4.8%)		
GnRH agonist	4 (13.8%)	2 (9.5%)		
Ovitrelle + GnRH agonist	20 (69%)	18 (85.7%)		
Endometrial thickness (mm)	9.7 ±1.9	10.1 ±2.5	0.55	
Number of follicles (>14 mm)	8.4 ± 4.6	7.9 ± 3.4	0.69	
Number of oocytes collected	13 ± 7.5	14 ± 7.5	0.62	
Number of M2 oocytes	11.14 ±7.4	11.29 ±6.8	0.94	
Number of fertilized oocytes (2PN)	8.07 ±6.6	8.05 ±5.6	0.99	
Usable embryos	3.7 ± 2.6	3.7 ± 2.7	0.92	
Number of top-quality embryos D3 + D5	3 ± 3.6	3 ± 2.8	0.94	
KID score	4.25 ± 1.26	4.68 ± 0.749	0.192	
Frozen embryos transferred D3 + D5	0.65 ± 1	0.50 ± 0.7	0.62	
Chemical pregnancy				
Fresh embryo transfer	15 (65.2%)	7 (50%)	0.49	
Frozen embryo transfer	9 (69.2%)	8 (61.5%)	1.0	
Clinical pregnancy				
Fresh embryo transfer	11 (47.8%)	4 (28.6%)	0.31	
Frozen embryo transfer	9 (69.2%)	7 (53.8%)	0.69	

* Values are mean ± SD or n (%).

**Table 3 cells-13-01872-t003:** Parameters reflecting proteins and mtDNA in follicular fluid, blood chemistry, and follicular fluid according to BMI.

Characteristic	BMI < 24.9	BMI > 25	*p*-Value	95% CI
Blood chemistry on OPU day				
Chemistry on OPU day in follicular fluid				
C-reactive protein (mg/L)	1.5 ± 1.6	5.2 ± 3.8	0.002	[−5.82, −1.61]
Cholesterol (mg/dL ± SD)	31 ± 35	23 ± 8.6	0.4	[−0.07, 25.40]
Triglycerides (mg/dL ± SD)	23 ± 8.6	14.7 ± 7.1	0.63	
HDL (mg/dL ± SD)	34 ± 26	20.9 ± 7.2	0.05	
LDL (mg/dL ± SD)	10 ± 29	11 ± 28.6	0.99	
BMP-15 concentration in FF (ng/mL)	34.8 ± 26.2	15.1 ± 10.5	0.006	[10.24, 38.70]
HSPG2 concentration in FF (ng/mL)	360.8 ± 119.2	440.6 ± 271	0.264	
mtDNA expression in CC (fold change)	1.10 ± 0.3	0.87 ± 0.18	0.016	[0.04, 0.41]

BMP-15—bone morphogenetic protein 15, HSPG2—heparan sulfate proteoglycan 2, mtDNA—mitochondrial DNA.

## Data Availability

Data are contained within the article.

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
