# Peer review of "Excess Weight Impairs Oocyte Quality, as Reflected by mtDNA and BMP-15"

_cells, 2024, doi:10.3390/cells13221872_

Round 1

Reviewer 1 Report

Comments and Suggestions for Authors

Sigal et al 2024 - Reviewers Comments

Overview

The authors have examined differences in hormone levels, lipid profiles, C-reactive protein and HSPG2 in the blood and follicular fluid between women with BMI above or below 25. They also compared BMP-15 levels in follicular fluid and mtDNA in cumulus cells from follicles from these women undergoing assisted reproduction as well as clinical pregnacy rates following IVF or ICSI to determine the effect of BMI on oocyte quality. The results of this study confirm that HDL is lower and LDL is higher in women that are overweight and the markers of oocyte quality (BMP-15 and mtDNA) were lower in follicular fluid and cumulus cells from these women. Clinical pregnancy rates did not differ. This appears to be a well designed study and a well-written manuscript with interesting results. To validate the results however, further detail of the methods should be provided and other suggested improvements are outlined below.

Title

Since the groups were separated into normal and “overweight” BMI, rather than obese you should avoid using the term obese or obesity in the title and when discussing the results.

Introduction

It would be helpful to include some information on CRP and HDL vs LDL in relation to BMI and oocyte quality in the introduction. Perhaps move some of the text from the discussion to the introduction section.

Lines 57-58 states “using two main factors...” but lists three; mtDNA, BMP15 and HSPG2.

Methods

Was there ethics approval required and obtained for this study?

Please describe the negative controls included in the mtDNA rtQPCR.

To validate the ELISA data it is important that more information is provided, including the inter- and intra-assay coefficient of variation and the linearity of the standard curves.

Line 133 Can you please clarify here that you measured cumulus cell mtDNA and FF BMP-15?

It is not clear if the mtDNA, and BMP-15 and HSPG2 in the FF were measured from individual follicles from each patient, and whether all follicles or just those with oocytes that were successfully fertilized and used for ET or frozen. Were single embryos transferred or multiple?

There are no methods provided for the hormone and lipid data.

Results

Table 3: What are the units for the mtDNA numbers in the table?

Figure 1: Please use different shaped symbols or filled and not filled circles as the colors used for each group are difficult to separate.

Figure 2: Please state what the lines of the boxes and error bars in the graph represent. It is also not clear what the fold-change is relative to and how many cumulus cell samples are represented in this graph.

Discussion

Line 235: Can you comment on how the BMP-15 levels in other studies compare to this study and why they might differ?

It would also be interesting to discuss what you think the mechanism is that links the lipid profile of overweight women to oocyte quality, with respect to the decreased BMP-15 secretion and mtDNA copy number.

Other minor comments

Check all abbreviations are written in full the first time eg. BMP-15, LDL, FET in the abstract and AFC in results section.

Author Response

Thank you very much for taking the time to review this manuscript. Please find the detailed responses below in the attached Word file. 

Reviewer 2 Report

Comments and Suggestions for Authors

The authors aimed to investigate the effects of obesity on the oocyte quality in the women of 18- 40 years. They mainly concluded from the mtDNA expression in cumulus cells (CC), and BMP-15 and Heparan sulfate proteoglycan 12 2 (HSPG2) in follicular fluid (FF).

This is an interesting topic. The authors may think about the BMI, not obesity, impairing oocyte quality. A higher or lower BMI is detrimental to oocytes.

Make BMP-15 and HSPG2 into the Boxplot as in Fig 2.  

Give more information about the BMI number (> 25 as obesity). This number is tricky. Some believe >30 is obesity. It is better to divide into four groups: lean, standard, overweight and obesity. If “lean” also impairs oocyte quality? Maybe we can get the optimal BMI for oocyte quality.   

Comments on the Quality of English Language

The authors aimed to investigate the effects of obesity on the oocyte quality in the women of 18- 40 years. They mainly concluded from the mtDNA expression in cumulus cells (CC), and BMP-15 and Heparan sulfate proteoglycan 12 2 (HSPG2) in follicular fluid (FF).

This is an interesting topic. The authors may think about the BMI, not obesity, impairing oocyte quality. A higher or lower BMI is detrimental to oocytes.

Make BMP-15 and HSPG2 into the Boxplot as in Fig 2.  

Give more information about the BMI number (> 25 as obesity). This number is tricky. Some believe >30 is obesity. It is better to divide into four groups: lean, standard, overweight and obesity. If “lean” also impairs oocyte quality? Maybe we can get the optimal BMI for oocyte quality.   

Author Response

Thank you very much for taking the time to review this manuscript. Please find the detailed responses  in the attached below Word file. 

Round 2

Reviewer 2 Report

Comments and Suggestions for Authors

No more comments